# Understanding the long-term impact of flooding on the wellbeing of residents: A mixed methods study

**Maureen Twiddy**[1,2☯]*, **Brendan Trump**[3☯], **Samuel Ramsden**[4☯]

**1** Hull York Medical School, University of Hull, Hull, United Kingdom, **2** Institute of Clinical and Applied Health Research, University of Hull, Hull, United Kingdom, **3** Hull York Medical School, University of York, Heslington, United Kingdom, **4** Flood Innovation Centre Energy and Environment Institute, University of Hull, Hull, United Kingdom

☯ These authors contributed equally to this work.

* m.twiddy@hull.ac.uk

**Data Availability Statement:** We have provided a minimal dataset as a supporting information file. Additional information cannot be provided as it risks identifying individuals. Therefore, information at ward level has been removed.

## Abstract

As the effects of climate change become more visible, extreme weather events are becoming more common. The effects of flooding on health are understood but the long-term impact on the well-being of those affected need to be considered. This mixed methods secondary analysis of a cross-sectional survey examined the extent to which being flooded in the past is associated with ongoing concerns about flooding. Survey data were collected from residents in Hull 11 years after the initial flooding event. Respondents were asked about the floods in 2007 and their current level of concern about flooding. Ordinal logistic regression explored the effect of age and tenancy status as predictors of current concern. Textual data were analyzed using thematic content analysis. Responses were received from 457 households, of whom 202 (48%) were affected by flooding in 2007. A fifth of respondents were very concerned about future flooding. Those who were not flooded were significantly less concerned about the risk of future flooding (U = 33391.0, z = 5.89, p < 0.001). Those who reported negative health and wellbeing effects from the floods were significantly more concerned about future flooding than those whose health was not affected (U = 7830.5, z = 4.43, p < 0.001). Whilst some residents were reassured by the introduction of new flood alleviation schemes, others did not feel these were adequate, and worried about the impact of climate change. The financial and emotional impacts of the floods still resonated with families 11 years after the event, with many fearing they would not cope if it happened again. Despite the 2007 floods in Hull happening over a decade ago, many of those affected continue to experience high levels of anxiety when storms are forecast. Residents feel powerless to protect themselves, and many remain unconvinced by the presence of new flood alleviation schemes. However, with the ongoing threat of climate change, it may be that other residents are unrealistic in their expectation to be 'protected' from flood events. Therefore, public health agencies need to be able to mobilize organizations to come together to pro-actively support families affected by flooding, to ensure those in need do not fall through the gaps of public healthcare delivery.

**Funding:** SR received an award from Living with Water to undertake the original Living with Water Project (no grant number). The funders have played no role in the study design, data collection and analysis, decision to publish or preparation of the manuscript.

**Competing interests:** The authors have declared that no competing interests exist.

# Introduction

As the effects of climate change become more visible, extreme weather events are becoming more common. These weather events bring natural disasters [1,2], most commonly flooding [3] with twenty-four percent of the global population exposed to floods—an increase of 4% (58–86 million people) from 2000 to 2015 [4]. One in six properties in England are at risk of flooding and 36,000 individuals were affected in the winter floods of 2015/16 alone [5].

The link between flooding and its impact upon mental health, particularly in regard to depression, anxiety and post-traumatic stress disorder are well established [6–9]. Some studies also highlight links with physical health [10,11] with effects ranging from gastroenteritis from waterborne pathogens to falls occurring during the aftermath. However, these studies only investigate the short-term effects, assessing individuals in the months or a few years after the index event. These studies show continuing health effects, so there remains a need to examine the long-term impacts on the communities affected, to improve our understanding of whether concerns regarding flooding remain in the minds of individuals who have previously been flooded, helping guide future healthcare plans.

Hull is a city in the North of England which is susceptible to flooding due to low lying land and location on the coast alongside a tidal river [12]. In 2007, Hull was hit by heavy rainfall causing severe surface level flooding resulting in major disruption across the city and tragically, loss of life [13]. It is reported that 20,000 people were affected by the 2007 floods with 8,600 households flooded [12]. In 2018, in a collaboration between academia, local government, and national and regional Flood Risk Management agencies (RMAs), a survey was conducted to understand communities' experiences of the 2007 flooding and how they felt about flooding now. This paper provides a secondary data analysis of the results of a 2018 survey (https://www.hull.ac.uk/editor-assets/docs/hull-household-flooding-survey-final-report.pdf) which asked participants about their experiences of the 2007 floods and about a smaller, subsequent flood which occurred in 2013. The survey asked respondents about a range of impacts the floods had on their lives, the help they received following the floods, and measures undertaken to reduce future impacts.

The objectives of this study are to investigate the extent to which prior experience of being flooded, and the impact on health and wellbeing, are associated with concerns about flooding a decade later, to help understand the lasting effects of these traumatic events on the individual.

# Method

## Study design and participants

This mixed methods secondary analysis was performed on data collected in the 2018 survey. This re-analysis examines data not previously reported. Ethical approval was obtained from the University of Hull, (Living with Water Project (Project 716042: Living H20 Socio-Impact Assessment). Ref No. 20172018563 Date: 11/09/2018.

Data were collected via a door-to-door survey conducted in three council wards in Hull; Beverley and Newland, Derringham and North Carr in 2018, with an online option to widen recruitment. All participants were informed about the purpose of the study and their right to withdraw at any time, and they provided verbal informed consent which was documented by the researcher for face-to-face data collection. For online respondents, an introduction to the survey was provided and consent was implied if they chose to complete the survey. All respondents were adults, minors under the age of 18 were excluded. The survey used a purposive sampling approach, specifically targeting streets known to have been flooded in 2007 and

asking participants about their experiences with the 2007 and 2013 floods. One survey was completed for each household surveyed (see supplemental data for survey questions). Derringham was the most severely affected by the 2007 flooding out of the three wards. Beverley and Newland is the most ethnically diverse of the wards and also has a high population of students. North Carr is the most deprived of the three wards, however there are pockets of deprivation within all three wards with Hull being identified as the 4th most deprived local authority in England and Wales [14].

Survey respondents were asked about their exposure to the 2007 floods and were categorized as: flooded house, flooded garden (exposure to flooding did not extend to inside the property), or not flooded, based on respondents' answers to the questions: '*how affected by 2007 floods'*. When grouping was unclear, analysis of the free text data was undertaken to determine flooding status. Demographic information was collected including age group, sex, ethnicity, residence status (owner/renter), employment status.

A single question was used to assess health effects: '*affected health and wellbeing'* (yes/no), and a follow-up open text question asked respondents to expand on their answer and explain *how* the flooding had affected their health and wellbeing. This provided respondents the opportunity to describe their health problems in more detail, and for specific health effects to be identified [15]. A single question was used, rather than a validated mental and wellbeing measure as the overall survey asked about a range of impacts from the floods, and the use of longer measures would have increased respondent burden. Respondents were recorded as having mental health effects if they self-reported experiencing stress, anxiety, or depression. Respondents were recorded as experiencing physical effects if they reported any physical health issue that they personally attributed to the floods (e.g. asthma exacerbations, rashes and falls). When respondents reported that both their physical and mental health were affected, they were assigned to both groups.

The quantitative outcome of interest in this reanalysis was self-reported concern about future flooding, which was assessed through one question '*level of concern about future flooding'*. Respondents answered on a 5-point Likert scale (1 = 'no concern', to 5 = 'very concerned'). Respondents who did not answer this question were excluded from this analysis. Respondents were also given the opportunity to explain their answer using an open text box, and the answers to this question were analyzed qualitatively.

## Data analysis

Statistical data analysis was undertaken using SPSS statistical software [16]. Descriptive statistics were calculated for all respondents, including those who did not experience flooding in 2007, with mean/SD reported, and median/IQR reported where data were not normally distributed. To assess the impact of self-reported physical and mental health effects of flooding in 2007 on concerns about future flooding risks in 2018, new categorical variables were calculated using the data extracted from the free-text boxes to classify respondents as reporting mental, physical or no health effects.

Ordinal regression was then used to predict level of ongoing concerns about future flooding. A Shapiro-Wilk test of normality was performed on the outcome variable (level of concern). This revealed a distribution of scores skewed towards higher values. Therefore, Mann-Whitney U tests were performed comparing groups. Ordinal regression examined the effect of age and tenancy status (owned/rented) on level of concern scores. Due to the small sample sizes, other theoretically relevant predictors such as ethnicity and working status were not included in the analysis. Gender was not included in the analysis due to in-person interviews often including both male and female parties.

Level of concern about future flooding scores were further classified as high concern (scores of 4–5) and moderate or low concern (scores 1–3) allowing comparison using an odds ratio between populations.

Open text box responses were subjected to thematic content analysis [17] to investigate respondents' answers to the questions "*why concerned about future flooding*" and "*worst part of the 2007 floods*". All data were read and re-read by two authors (BF and MT) and an initial coding frame developed to classify topics discussed by respondents. This coding frame was applied to all responses by BF in Excel [18].

## Results

Surveys were sent to a total of 2000 unique households of which 457 were returned providing sufficient data for analysis (154 respondents answered online; 303 answered in person during door to door interviews). A response rate of 15% from target wards was obtained, excluding online responses. Of the 457 respondents, just over half were not flooded (n = 235), 222 (48.6%) experienced some flooding (159 (34.8%) reported their house flooded and 63 (13.8%) experienced flooding into their garden). Of those responding to the survey, 259 (57%) had been resident since before the 2007 floods; 171 (66%) of whom were affected by flooding in 2007 (in their current home); 198 (43%) had moved into their present home after the 2007 floods, of whom 51 (26%) were flooded in 2007 (previous home flooded).

Of the respondents whose homes were flooded in 2007, 124 (55.9%) were female, 169 (76.1%) were owner occupiers and 38.3% were over 65 years old (Table 1), indicating that respondents who were flooded have a different demographic profile to the larger community.

### Health and wellbeing

Ninety (40.5%) of the 222 who reported flooding to their home or garden reported that their health and wellbeing had been affected by the floods, with 75 (33.8%) reporting effects on their mental health (e.g. anxiety or depression), and 24 (10.8%) reporting physical health effects (e.g. trips and falls), with a small proportion (n = 9) reporting both physical and mental health effects.

### Concern about future flooding

Twenty percent of all respondents are very concerned about flooding, with concern highest for people who were flooded. Scores for '*level of concern regarding future flooding*', were higher for those who had experienced flooding in 2007 (median = 3, IQR = 3, mean = 3.41, SD = 1.29) compared to those who were not flooded in 2007 (median = 2, IQR = 3, mean = 2.63, SD = 1.41) (Fig 1). There was little difference in concern regarding future flooding scores for those whose homes were flooded (median = 3, IQR = 3, mean = 3.40, SD = 1.34), and for those whose gardens were flooded (median = 4, IQR = 2, mean = 3.44, SD = 1.18).

A Mann-Whitney U test, indicated that the level of concern regarding future flooding was greater for those who were flooded in 2007 (Mdn = 3) than for those who were not flooded (Mdn = 2) (U (flooded n = 220, not flooded n = 231) = 33391, z = 5.89, p < 0.001). Therefore, subsequent analyses focused on those who were flooded.

As shown in Fig 2, concern about future flooding for those who reported negative health and wellbeing effects from flooding in 2007 (Mdn = 4, IQR = 2, Mean = 3.85) differed significantly from those who were flooded but did not report any negative health effects (Mdn = 3, IQR = 2, mean = 3.11) (U (Health affected n = 89, Health not affected n = 131) = 7830.5, z = 4.43, p < 0.001). This effect size was also found to be moderate with r = 0.3.

Level of concern scores were then dichotomized to high concern (scores of 4 or 5) and moderate/low concern (scores of 1 to 3). Sixty-four percent of those who reported negative

**Table 1. Sample demographics.**

| Group | | Whole sample N (%) | Only those affected by Flooding N (%) |
|---|---|---|---|
| **Ward** | B & N | 119 (26.1) | 38 (17.1) |
| | Derringham | 169 (37) | 120 (54.1) |
| | North Carr | 62 (13.6) | 9 (4.1) |
| | Other | 107 (23.4) | 55 (24.8) |
| **Own or rent property** | Own | 291 (63.7) | 169 (76.1) |
| | Rent | 166 (36.3) | 53 (23.9) |
| **Flooding level** | Not flooded | 235 (51.4) | |
| | House flooded | 159 (34.8) | 159 (71.6) |
| | Garden flooded | 63 (13.8) | 63 (28.4) |
| **Affected health and wellbeing** | Yes affected | 92 (20.1) | 90 (40.5) |
| | No, not affected | 366 (80.1) | 132 (59.5) |
| | Mental health effect | 77 (16.8) | 75 (33.8) |
| | Physical effect | 24 (5.3) | 24 (10.8) |
| **Gender** | Couple | 5 (1.1) | 3 (1.4) |
| | Female | 247 (54) | 124 (55.9) |
| | Male | 193 (42.2) | 88 (39.6) |
| | Prefer not to say | 12 (2.6) | 7 (3.2) |
| **Age group** | 18–24 | 43 (9.4) | 10 (4.5) |
| | 25–34 | 64 (14) | 13 (5.9) |
| | 35–50 | 98 (21.4) | 48 (21.6) |
| | 51–64 | 106 (23.2) | 65 (29.3) |
| | 65–79 | 108 (23.6) | 64 (28.8) |
| | 80+ | 34 (7.4) | 21 (9.5) |
| **Ethnicity** | White British | 400 (87.5) | 209 (94.1) |
| | Other white background | 18 (3.9) | 2 (0.9) |
| | African | 5 (1.1) | 1 (0.5) |
| | Arab | 1 (0.2) | 0 |
| | Chinese | 3 (0.7) | 0 |
| | Pakistani | 2 (0.4) | 0 |
| | Indian | 2 (0.4) | 0 |
| | Other Asian background | 4 (0.9) | 1 (0.5) |
| | Mixed ethnic background | 6 (1.3) | 1 (0.5) |
| | Other ethnic group | 2 (0.4) | 1 (0.5) |
| | Prefer not to say | 14 (3.1) | 7 (3.2) |
| **Employment status** | Employed | 182 (39.8) | 78 (35.2) |
| | Caring for relatives | 9 (2) | 2 (0.9) |
| | Retired | 158 (34.6) | 97 (43.7) |
| | Self employed | 32 (7) | 20 (9) |
| | Student | 20 (4.4) | 2 (0.9) |
| | Volunteering | 1 (0.2) | 0 |
| | Out of work | 28 (6.1) | 12 (5.4) |
| | Other | 19 (4.2) | 8 (3.6) |
| | Prefer not to say | 8 (1.8) | 3 (1.4) |
| **Total** | | 457 | 222 (48.6) |

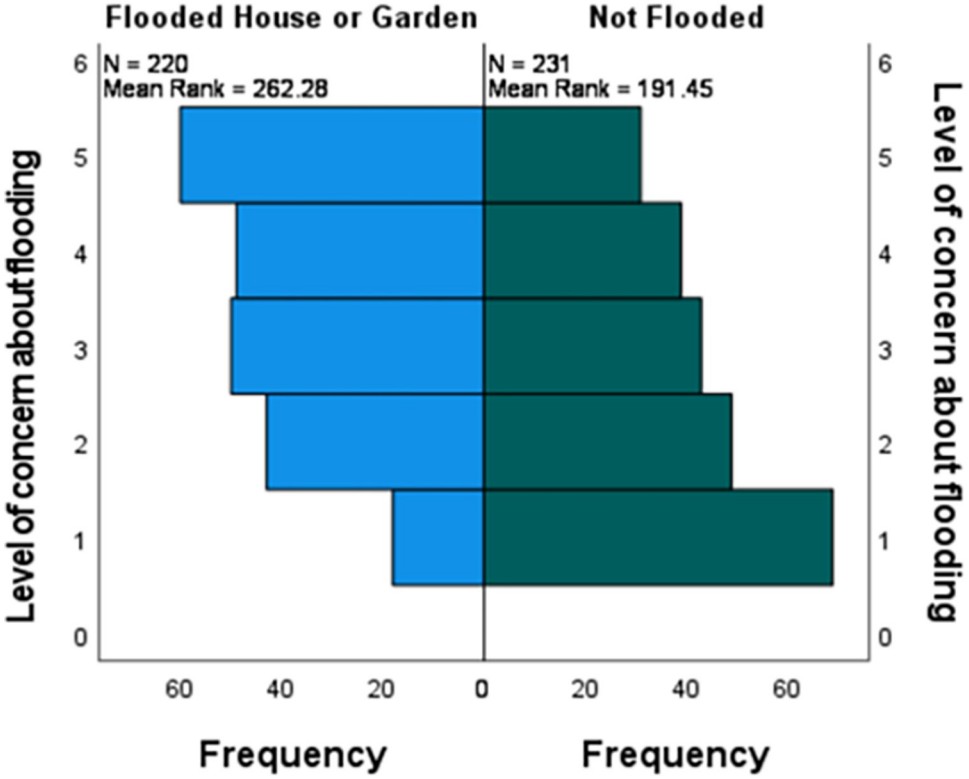

**Fig 1. Level of concern about future flooding scores for those flooded in 2007 compared to those who were not flooded.**

health and wellbeing effects from the 2007 floods reported high levels of concern compared to 39.7% of those who reported no health and wellbeing effects from the floods. Concern about future flooding was associated with reporting previous negative health and wellbeing effects (OR 2.71 95% CI 1.55–4.72).

Those who reported mental health effects were more concerned about future flooding (mean = 4.05, SD 1.18, n = 76) compared to those reporting physical health effects (mean = 3.83, SD 1.24, n = 24). Further comparisons using these groups were not performed due to the small sample sizes present.

### Predictors of concern about future flooding

The results of an ordinal regression analysis, (Table 2), show the effect of age and tenancy status (owned vs rented) on level of concern about future flooding, for those flooded in 2007. For homeowners, the odds of being concerned over risk of flooding was 1.32 (CI 95%, 0.91 to 1.90) compared to those who rented (Wald $X^2$ (df = 2.16, p = 0.14).

The odds of reporting higher levels of concern about future flooding show that compared to those aged over 80 years, respondents aged 35–50 years had the highest odds of reporting concerns OR 3.96 (95% CI, 1.95 to 8.06); Wald $X^2$ (df = 14.42, p < 0.0001).

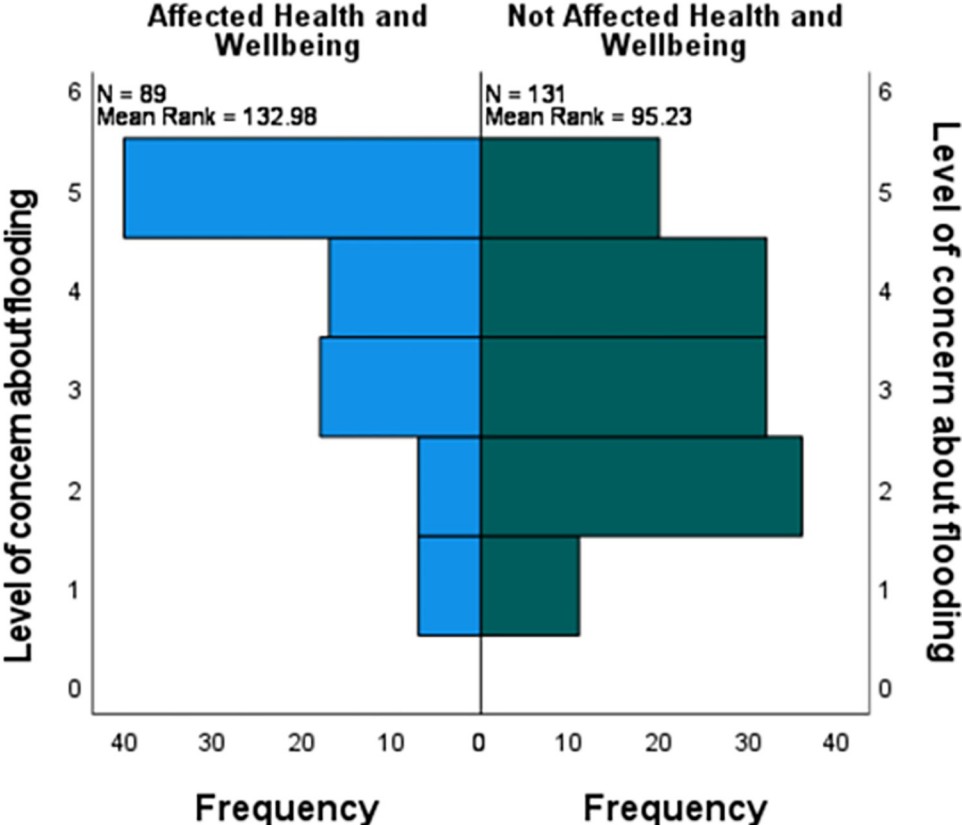

**Fig 2. Level of Concern about future flooding scores for those who were flooded, comparing those whose health and wellbeing was negatively affected to those health and wellbeing was not affected.**

## Qualitative findings

Analysis of the quantitative data found that people who had experienced flooding in 2007 varied in their level of concern about future flooding. The qualitative data provided a better understanding of these patterns. A free text box was provided to allow respondents the

**Table 2. Regression analysis results (those affected by flooding in 2007).**

| Variable | Wald | Significance | Std Error | Odds Ratio | Lower 95%CI | Upper 95% CI |
|---|---|---|---|---|---|---|
| **18–24 years** | 0.36 | 0.55 | 0.43 | 1.29 | 0.56 | 2.96 |
| **25-34- years** | 2.91 | 0.09 | 0.39 | 1.93 | 0.91 | 4.12 |
| **35–50 years** | 14.42 | 0.0001 | 0.36 | 3.96 | 1.95 | 8.06 |
| **51–64 years** | 9.99 | 0.002 | 0.36 | 3.07 | 1.53 | 6.15 |
| **65–79 years** | 3.74 | 0.05 | 0.36 | 1.97 | 0.99 | 3.96 |
| **80+ years** | | | | 1.00 | | |
| **Own** | 2.16 | 0.14 | 0.19 | 1.32 | 0.91 | 1.9 |
| **Rent** | | | | 1.00 | | |

opportunity to explain why they were, or were not, concerned about future flooding. Responses varied in length from a few words, to a paragraph. Our analysis included only those who reported being flooded, either in their garden or home in 2007. Of the 159 people whose house was flooded 152 provided textual data for analysis. Of the 63 whose garden was affected by flooding, 60 provided textual data for analysis.

Some people expressed low concern about future flooding and this was explained by their belief that the flooding was a 'freak event' and so unlikely to reoccur, and a belief that any residual risk would be managed by the presence of new flood alleviation systems. For those who remain concerned, explanations are linked to concerns about flooding due to climate change, the limitations of flood alleviation schemes and fears about the effects future flooding would have on their quality of life.

**Theme 1: Flooding as a 'freak event'.**   Older residents in particular were likely to attribute the flooding to a freak event and therefore unlikely to happen again. Some had lived in their homes for many years prior to and since the flooding, and put the risk into that context. Others put the risk of flooding out of their minds because it was something that was out of their control.

*"2007 was a freak event. This area is not normally prone to flooding. I have lived here for 64 years." Female 80+*

*"I am not concerned because I can't do anything about the floods." Male 80+*

*"It still bothers me that it could happen again but family reassure me that it is a one in a life-time thing as it has never happened before." female, 35–50*

**Theme 2: Climate change and extreme weather.**   In contrast to those who attributed the flooding to being a freak event, the majority of those who were flooded in 2007 felt their risk from flooding was ever present. A commonly held view across age groups was that continuing climate change may make existing problems worse, with 10 people specifically describing their concerns in the context of climate change, with others alluding to it.

*"Hull is situated below sea level and is prone to flash flooding after heavy rainfall. With climate change, extreme weather conditions are predicted to become more frequent." Male 25–34*

*"We need to look after our property. Getting older makes it 'scary' especially with global warming." 65–79 Female*

All of those who were most concerned about future flooding talked about how heavy rainfall left them anxious, with one lady describing a feeling of panic with heavy rainfall. Others used terms such as scared, panicky or concerned, and described how a forecast of heavy rainfall would lead them to actively prepare themselves for the worst, for example, by moving treasured possessions upstairs, blocking air bricks and watching the street for signs of flooding.

*"I just panic all the time when there is a heavy rainfall" 18–24 Female*

*"Whenever it rains now we all start to prepare ourselves, no matter how bad it is forecast" Male 18–24*

**Theme 3: Flood alleviation.**   Flood alleviation schemes had done much to quell the fears of some residents. Being able to observe tangible changes such as the development of new

flood alleviation schemes and drainage systems led to confidence in the ability of Flood Risk Management Agencies to manage future floods. For some, there was a strong belief that this work had alleviated the risk of flooding, with 63 respondents talking positively about flood alleviation schemes. However, a few questioned whether these were sufficient to protect their homes from future flooding, and some thought that whilst other areas had been targeted for new flood defenses, the areas in which they live have been disregarded.

*"there has been a lot of work around the city" Male 80+*

*"Whilst flood lagoons and flood alleviation has been provided to west of Derringham, there is still flooding on Setting Dyke fields" Male 51–64*

A large proportion of those who had previously been flooded remained concerned about flooding, citing a lack of clearance to local drains as a major risk factor. These residents feared that drains would get blocked and poor maintenance of drainage systems such as ditches meant they were still at risk of future flooding due to poor land drainage.

*"It's always in my mind when rain is very heavy and the drain outside my house is blocked again" Female 51–64*

The development of new housing, especially on green field sites made residents question whether new housing developments on flood plains were a good idea.

*"I am concerned that the excessive amount of new housing is putting greater pressure on the drains and infrastructure of the village. Roads and certain areas are experiencing flash floods on a too regular basis." Female 51–64*

**Theme 4: Effects of future flooding on life.** A persistent topic related to financial aspects of flooding, with 17 residents specifically discussing the cost of house insurance and how previous flooding had increased the flood risk rating from insurers, raising their premiums, and making the cost of insurance inaccessible to some. For these families, future flooding would be a disaster as they would not be able to make repairs to their homes.

*"I can't go through it again and the insurance will cripple us" Male 35–50*

*"Worried that insurance wouldn't pay out in a house that has previously flooded. Worried for my disabled child's health and the fact that he often needs emergency ambulance access for his epilepsy." (Female 25–50)*

In contrast, a few felt they were better prepared for future floods because they had invested in good home insurance and they, and the council had made improvements to make their home flood secure.

*"I have good home insurance, I don't skimp on this. The Council [. . .] have put in flood defenses at the end of the gardens that border the field [. . .] In 2013 the flood defenses worked [. . .] I have lots of trees, shrubs and plants in the garden. . ." Female 51–64*

The fear of losing treasured possessions meant that some would not leave anything of value downstairs in case of any flooding in the future.

*"if it happens again I'm terrified of losing everything I've scrimped and saved for and daren't leave anything sentimental downstairs"* Gender Unspecified 25–34

A common concern was the impact that future flooding might have on the health and well-being of their children and vulnerable family members. Stories about how past flooding affected access to healthcare and schooling were common, and fears about how future flooding might impose upon a child's life were continuing concerns.

*"Worried for my disabled child's health and the fact that he often needs emergency ambulance access for his epilepsy."* Female 35–50

*"The sheer disruption and upset of my children ages 7,6,3 and a 10 week premature baby we basically lost everything downstairs including all their toys"* Female 35–50

## Discussion

The results of the quantitative analysis revealed significant differences in levels of concern about future flooding between those who were flooded and those not flooded despite it being over a decade since the initial flooding event. Many members of the community were nervous at the prospect of heavy rain and some took evasive action when rain threatened. The level of concern expressed was similar for those whose home was flooded and those where only their garden was flooded. The quantitative data indicates that those who were flooded in 2007 and reported that this had a detrimental effect upon their health and wellbeing were more likely to be concerned about future flooding than those who reported no health impacts. However, the control group for these analyses were respondents who were not flooded, and a matched sample of flooded and non-flooded respondents could not be formed; the flooded population are slightly older, and more likely to be owner-occupiers, and more likely to be female, which limits our ability to firm strong conclusions about cause and effect. However, our findings support earlier research which suggests flooding causes widespread disruption to life and services impacting mental health [19].

Previous studies have found that populations exposed to floods have a high prevalence of common mental health problems, with effects greatest in those with the lowest social capital [9,20]. These studies show the physical and financial impacts of flooding are associated with poorer mental health, and having to relocate during the flooding clear up is a strong predictor of negative mental health outcomes. Studies also show that mental health morbidity exists up to five years post exposure [8,20,21]. The present study adds to this evidence as residents in Hull still reported flood related anxiety over a decade after the index event; although limitations to the study design must be acknowledged. One interpretation of our findings is that people who are already anxious misattribute the cause of their anxiety to the flooding–due to the study design this cannot be explored further. A further limitation of the present study is that mental health impact was not assessed using validated measures. Nevertheless, our findings suggest this area warrants further investigation.

The regression analysis examining possible explanations for concerns about future flooding, in those who had previously experienced flooding in 2007 found that those aged between 35 and 64 years were most concerned, with these groups reporting significantly higher concern scores when compared to individuals aged over 80 years. Low concern scores were reported by young adults which may be an indication of psychological resilience to the impact of extreme weather [22] but could also be explained by them being children during the floods with their

parents having to deal with its repercussions. Our qualitative findings explain some of our quantitative findings. Those residents currently aged between 35 and 64 years of age are most likely to have family responsibilities, and concern for the wellbeing of family members was described repeatedly by these respondents, with few having the resources to reduce their personal vulnerability to flooding. The qualitative data also found that elderly respondents were more likely to see the flooding as a 'freak event' which had occurred only once or twice during their lifetime. One explanation is that having coped with the previous floods they were less worried about it happening again, a finding that has been reported elsewhere [22], although the opposite finding has also been reported [23], suggesting context and severity are likely to be important.

We hypothesized that house ownership would influence flooding concerns, but although the odds ratio was higher, this was not statistically significant. This may be because regardless of ownership, people were still forced out of their homes and were disrupted by the floods. However, we did find that people who rented their homes were less likely to possess adequate contents insurance which added to their concerns [24]. The qualitative results also highlighted issues faced by families trying to secure adequate home insurance after they were flooded, with the economic and social consequences of flooding singled out as reasons for their continued fear of flooding. This echoes the findings of Bang and Church Burton [25], who found that although flood insurance is needed for a mortgage it is only guaranteed for properties with a low flood probability. It is therefore unsurprising that residents living in an area of high urban deprivation will be concerned about the risk of future flooding, as some will be unable to afford the financial consequences of another flood.

The qualitative data revealed that many residents felt not enough has been done by RMAs to protect them from flooding. Perceptions about the (in)ability of the flood alleviation scheme to cope with future floods, a lack of maintenance of drains and dykes and a lack of any apparent coverage in some areas, all contributed to concerns. Similarly, building of new homes on flood plains were seen as reducing the amount of land available to absorb future flood water. These have implications for the trust and confidence that communities have in the regional flood risk management agencies. Studies suggests that there is a higher likelihood that individuals who experience disasters perceive themselves to be more vulnerable to climate risks, and the sense of helpless reported in the present study reflects this [25–27]. It must however be noted that not everyone who was flooded in 2007 felt they were at high risk of future flooding. Risk can be perceived in the context of being prepared, the perceived probability, or likely consequences of the event [28], and this reflects the views of some respondents.

## Strengths and limitations

A limitation of the present study is its retrospective nature. Respondents were asked to recall the impact of flooding a decade earlier, and so some health effects reported may not be directly related to the flooding. To attenuate this the analysis focused only on the broad concepts of mental and physical health, and on how respondents felt now about an objective experience (flooding), which reduces the risk of recall bias. However, it may also be the case that those individuals who are more anxious may be more likely to attribute their health problems to the floods than less anxious people. A matched sample would have allowed for this to be tested, but this could not be formed from the data available. A further limitation was that the study did not use validated measures of mental or physical health to assess these effects. Mental and physical health impacts of flooding were not the focus of the original survey, and so a decision was taken to minimize respondent burden, and rely on self-reported data.

A key strength of the study was the use of a purposive sampling approach using door-to-door interviews and online methods which maximized opportunities to participate. Although the door-to-door interviews took place during working hours, the addition of online methods allowed working age residents to respond, and this is reflected in the demographics of respondents. There is a possibility of self-selection bias in such surveys, with those greatest affected most likely to reply. However, our large sample size, and the distribution of data indicate that this was a topic of interest to the wider community, as half of respondents had not personally experienced flooding, but nevertheless had thoughts they wished to share.

## Application

This analysis adds to the growing evidence that demonstrates the need for long-term surveillance of those affected by flooding [9]. The data presented here suggest particular attention needs to be given to assessing the longer-term health and wellbeing of those affected by flooding events. The findings indicate that ongoing anxiety is not confined to those whose homes were directly impacted by flooding, and interventions are needed to improve community and individual resilience. Significantly more support is needed for communities where the financial stress of flooding is highest, and dissemination of information to local communities about improvements and continuing investment to alleviate upcoming pressures from the growing climate crisis needs to be improved.

Future studies are needed, using validated measures of mental and physical health, to establish how widespread long-term concerns about flooding are, and the relation between socio-economic deprivation and long term physical and mental health impacts, to inform the development of interventions to support communities.

## Conclusion

Despite the passage of time and improvements made to the flood defenses throughout Hull we have identified that concerns remain over a decade later. Residents who have been flooded in the past are most likely to be concerned about the effect a future flood would have on their lives, and their ability to recover from its effects. The impact of climate change and the regular flood events across the UK mean the threat is always in the news. Many residents feel powerless to protect themselves, and remain unconvinced that enough has been done enough to protect them. It may be that residents are unrealistic in their expectation to be 'protected' from flood events. This being the case, public health agencies need to ensure organizations are ready to come together to pro-actively support households affected by flooding, to ensure those in need do not fall through the gaps of public healthcare delivery.

## Supporting information

**S1 File. Final Hull LWW survey 2018.**
(PDF)

**S2 File. Dataset.**
(SAV)

## Acknowledgments

Thanks go to Yorkshire Water, Hull City Council, East Riding of Yorkshire Council, and the Environment Agency who were partners in this study. Thanks too, to the residents of Hull for their input into this survey.

## Author Contributions

**Conceptualization:** Maureen Twiddy, Samuel Ramsden.

**Formal analysis:** Maureen Twiddy, Brendan Trump.

**Funding acquisition:** Samuel Ramsden.

**Methodology:** Maureen Twiddy.

**Resources:** Samuel Ramsden.

**Supervision:** Maureen Twiddy, Samuel Ramsden.

**Writing – original draft:** Brendan Trump.

**Writing – review & editing:** Maureen Twiddy, Samuel Ramsden.

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
