## [Decision Letter · Decision Letter 0]

5 Aug 2022

PONE-D-22-11055Understanding the long-term impact of flooding on the wellbeing of residents: a mixed methods studyPLOS ONE

Dear Dr. Twiddy,

Thank you for submitting your manuscript to PLOS ONE. After careful consideration, we feel that it has merit but does not fully meet PLOS ONE’s publication criteria as it currently stands. Therefore, we invite you to submit a revised version of the manuscript that addresses the points raised during the review process.

ACADEMIC EDITOR:

Thank you for your submission. I agree with both reviewers, please see their comments below. Reviewer #2 highlights the importance of your current manuscript and Reviewer #1 highlights a few limitations in the methods and comparison group for your analyses.

Please look carefully at the comments and be responsive. If you are not able to create the comparison group that Reviewer #1 suggests, then you will need to enhance the limitations section with the concerns over the comparison group, as stated very clearly by Reviewer #2. In addition, discussing the implications of that limitation for the conclusions that can be drawn from your analyses.

We look forward to receiving your revised manuscript.

Kind regards,

Ali A. Weinstein, Ph.D.

Academic Editor

PLOS ONE

Journal Requirements:

Reviewers' comments:

Reviewer's Responses to Questions

**Comments to the Author**

1. Is the manuscript technically sound, and do the data support the conclusions?

Reviewer #1: Yes

Reviewer #2: Yes

2. Has the statistical analysis been performed appropriately and rigorously? 

Reviewer #1: No

Reviewer #2: Yes

3. Have the authors made all data underlying the findings in their manuscript fully available?

Reviewer #1: Yes

Reviewer #2: Yes

4. Is the manuscript presented in an intelligible fashion and written in standard English?

Reviewer #1: Yes

Reviewer #2: Yes

5. Review Comments to the Author

Reviewer #1: In this paper, the authors use both quantitative and qualitative methods to explore the impact of 2007 floods on residents of Hull, UK, based on a 2018 survey. The authors find that affected residents still experience significant levels of stress and anxiety around flooding. As climate hazards of all types become more prevalent, the long term physical and emotional effects of climate hazard exposures will mount and are worth studying. Thus, the analytical goals of this paper contribute to a problem that has a high and growing real world importance. The qualitative analysis highlights several key themes that Hull residents emphasized in their thinking about past and future flood risk. These are valuable for understanding local perspective and concerns

I have some concerns about the empirical support for the authors’ conclusions.

1. The study population is not well defined. It appears that other residents of the same geographic location, regardless of the duration of their residence in the study areas are the control group for flood affected residents. Over an 11 year period, there are likely to be significant population changes. Importantly, people who left may have been less vulnerable, healthier, or had other unmeasurable characteristics that mean their physical and mental health is systematically different they population of long term residents. This population shift means that we can’t make any systematic claims about people affected by the 2007 floods. Table 1 shows that the flood affected population is substantially different than the whole sample- it is older, white, more female, and more likely to own thier home. These are to be expected given a requirement of an 11 year residence in a single location. But it suggests that their neighbors’ are not a good control group.

a. If there is sufficient data, a matched sample approach is likely to be a better strategy for inference. For each flooded resident, search within the control group for a household with a similar demographic profile.

2. The self report about health effects caused by flooding, mental health effects, and concern about flooding is also likely to lead to conflation of cause and effect. An anxious person may be more likely to attribute their asthma to prior flooding, and also to report high levels of concern. This is a notable explanation for the association found in Fig 2.

3. Table 2 doesn’t report if it’s analyzing the flood affected population only, or the whole population.

Reviewer #2: This is an excellent paper that presents an assessment of the health and wellbeing impacts of floods more than a decade on from a major flood event. The results are of interest internationally and are well presented. I recommend acceptance of the paper with two very minor things to adjust which could be dealt with at proof stage. First, when mentioned please provide a link to formal scales for measuring health and wellbeing. Second there are some (three or four) misspellings in the discussion that need to be changed. Otherwise the paper is good to go for me. Well done on an excellent paper and a rare insight into an important aspect of the impacts of floods, years after.

6. PLOS authors have the option to publish the peer review history of their article (what does this mean?). If published, this will include your full peer review and any attached files.

Reviewer #1: No

Reviewer #2: No

---

## [Author Response · Author response to Decision Letter 0]

16 Aug 2022

Response to editorial and reviewer comments

Thank you for the opportunity to resubmit our article. We have made revisions based on the feedback from the two reviewers, and the requests made by the editor. I have responded to each comment below. For ease, I have included the text from the manuscript in my response and cited where in the manuscript the text can be found. The page and line numbers refer to the ‘manuscript with revisions’.

1. We have ensured that the manuscript now meets PLOS ONE's style requirements, and have used the guidelines to make revisions

2. We have added additional details regarding participant consent. These can be found on page 5, and text provided here for ease of review. New text is in italics - below. Minors were not eligible to participate, text to this effect is included. 

Data were collected via a door-to-door survey conducted in three council wards in Hull; Beverley and Newland, Derringham and North Carr in 2018, with an online option to widen recruitment. All participants were informed about the purpose of the study and their right to withdraw at any time, and they provided verbal informed consent which was documented by the researcher for face-to-face data collection. For online respondents, an introduction to the survey was provided and consent was implied if they chose to complete the survey. All respondents were adults, minors under the age of 18 were excluded. 

3. We are including our minimum dataset as a supplementary file as requested. All data were collected anonymously, but as data include personal information we have not provided a breakdown by ward as this risks identifying individuals. 

REVIEWER COMMENTS

We thank the reviewers for their positive comments, which have been helpful and have allowed us to improve the manuscript, and clarify the study design more clearly for readers. 

Reviewer #1: In this paper, the authors use both quantitative and qualitative methods to explore the impact of 2007 floods on residents of Hull, UK, based on a 2018 survey. The authors find that affected residents still experience significant levels of stress and anxiety around flooding. As climate hazards of all types become more prevalent, the long term physical and emotional effects of climate hazard exposures will mount and are worth studying. Thus, the analytical goals of this paper contribute to a problem that has a high and growing real world importance. The qualitative analysis highlights several key themes that Hull residents emphasized in their thinking about past and future flood risk. These are valuable for understanding local perspective and concerns

I have some concerns about the empirical support for the authors’ conclusions.

1. The study population is not well defined. It appears that other residents of the same geographic location, regardless of the duration of their residence in the study areas are the control group for flood affected residents. Over an 11 year period, there are likely to be significant population changes. Importantly, people who left may have been less vulnerable, healthier, or had other unmeasurable characteristics that mean their physical and mental health is systematically different they population of long term residents. This population shift means that we can’t make any systematic claims about people affected by the 2007 floods. Table 1 shows that the flood affected population is substantially different than the whole sample- it is older, white, more female, and more likely to own their home. These are to be expected given a requirement of an 11 year residence in a single location. But it suggests that their neighbors’ are not a good control group.

a. If there is sufficient data, a matched sample approach is likely to be a better strategy for inference. For each flooded resident, search within the control group for a household with a similar demographic profile.

2. The self report about health effects caused by flooding, mental health effects, and concern about flooding is also likely to lead to conflation of cause and effect. An anxious person may be more likely to attribute their asthma to prior flooding, and also to report high levels of concern. This is a notable explanation for the association found in Fig 2.

3. Table 2 doesn’t report if it’s analyzing the flood affected population only, or the whole population.

Response to Reviewer #1 

1. We have added more information to characterise the population (page 8 lined 158-160) – new text in italics. We are somewhat limited in what we can say about the population, as this is a secondary analysis of existing data, so only have the data available to us from the previous study. 

“Of the 457 respondents, just over half were not flooded (n=235), 222 (48.6%) experienced some flooding (159 (34.8%) reported their house flooded and 63 (13.8%) experienced flooding into their garden). Of those responding to the survey, 259 (57%) had been resident since before the 2007 floods; 171 (66%) of whom were affected by flooding in 2007 (in their current home); 198 (43%) had moved into their present home after the 2007 floods, of whom 51 (26%) were flooded in 2007 (previous home flooded

Of the respondents whose homes were flooded in 2007, 124 (55.9%) were female, 169 (76.1%) were owner occupiers and 38.3% were over 65 years old (Table 1), indicating that respondents who were flooded have a different demographic profile to the larger community.”

2. We recognise that residents who were not flooded do not make a good comparison group, but we were not able to produce a matched sample as residents who were not flooded were not asked some questions, making a matched sample impossible. 

We have provided more contextual data about the sample (see above). We have also expanded the discussion to acknowledge the limitations of our study design. New text in italics below.

Line 328-332: “The quantitative data indicates that those who were flooded in 2007 and reported that this had a detrimental effect upon their health and wellbeing were more likely to be concerned about future flooding than those who reported no health impacts. However, the control group for these analyses were respondents who were not flooded, and a matched sample of flooded and non-flooded respondents could not be formed; the flooded population are slightly older, and more likely to be owner-occupiers, and more likely to be female, which limits our ability to firm strong conclusions about cause and effect. However, our findings support earlier research which suggests flooding causes widespread disruption to life and services impacting mental health (19). “

Reviewer #1 also noted that self report about health effects caused by flooding, mental health effects, and concern about flooding is also likely to lead to conflation of cause and effect. 

We acknowledge this may be the case, but our findings are also in line with other studies, so do not feel this is the likely cause. However, to address the reviewers concerns we have included additional text in the discussion to address this issue (page 18/19 lines 341-345)

“The present study adds to this evidence as residents in Hull still reported flood related anxiety over a decade after the index event; although limitations to the study design must be acknowledged. One interpretation of our findings is that people who are already anxious misattribute the cause of their anxiety to the flooding – due to the study design this cannot be explored further. A further limitation of the present study is that mental health impact was not assessed using validated measures. Nevertheless, but our findings suggest this area warrants further investigation.”

We have revised the title for Table 2 to make it clear that the population is only those affected by flooding in 2007.

Reviewer #2: This is an excellent paper that presents an assessment of the health and wellbeing impacts of floods more than a decade on from a major flood event. The results are of interest internationally and are well presented. I recommend acceptance of the paper with two very minor things to adjust which could be dealt with at proof stage. First, when mentioned please provide a link to formal scales for measuring health and wellbeing. Second there are some (three or four) misspellings in the discussion that need to be changed. Otherwise the paper is good to go for me. Well done on an excellent paper and a rare insight into an important aspect of the impacts of floods, years after.

Response to Reviewer #2

Thank you for the very positive comments. We have identified and revised the typographical errors in the manuscript. 

We did not use a formal scale for measuring health and wellbeing. This was stated in the methods section, but to make this clearer, we have added additional text (page 6 lines 114-116): 

“A single question was used to assess health effects: ‘affected health and wellbeing’ (yes/no), and a follow-up open text question asked respondents to expand on their answer and explain how the flooding had affected their health and wellbeing. This provided respondents the opportunity to describe their health problems in more detail, and for specific health effects to be identified (15). A single question was used, rather than a validated mental and wellbeing measure as the overall survey asked about a range of impacts from the floods, and the use of longer measures would have increased respondent burden.”

---

## [Decision Letter · Decision Letter 1]

7 Sep 2022

Understanding the long-term impact of flooding on the wellbeing of residents: a mixed methods study

PONE-D-22-11055R1

Dear Dr. Twiddy,

We’re pleased to inform you that your manuscript has been judged scientifically suitable for publication and will be formally accepted for publication once it meets all outstanding technical requirements.

Kind regards,

Ali A. Weinstein, Ph.D.

Academic Editor

PLOS ONE

Reviewers' comments:

Reviewer's Responses to Questions

**Comments to the Author**

1. If the authors have adequately addressed your comments raised in a previous round of review and you feel that this manuscript is now acceptable for publication, you may indicate that here to bypass the “Comments to the Author” section, enter your conflict of interest statement in the “Confidential to Editor” section, and submit your "Accept" recommendation.

Reviewer #1: All comments have been addressed

Reviewer #2: All comments have been addressed

2. Is the manuscript technically sound, and do the data support the conclusions?

Reviewer #1: Yes

Reviewer #2: Yes

3. Has the statistical analysis been performed appropriately and rigorously? 

Reviewer #1: Yes

Reviewer #2: Yes

4. Have the authors made all data underlying the findings in their manuscript fully available?

Reviewer #1: Yes

Reviewer #2: Yes

5. Is the manuscript presented in an intelligible fashion and written in standard English?

Reviewer #1: Yes

Reviewer #2: Yes

6. Review Comments to the Author

Reviewer #1: Thank you for the thorough response to my empirical concerns. I understand the limitations of working with 'found' data. The revisions performed clearly articulate the limitations of the existing dataset, which prevent a more complete and rigorous statistical analysis. The authors are careful to explain the strengths and limitations of their analysis in the revised paper.

Reviewer #2: I am happy that the authors have addressed my comments and I am happy to accept the paper for publication

7. PLOS authors have the option to publish the peer review history of their article (what does this mean?). If published, this will include your full peer review and any attached files.

Reviewer #1: No

Reviewer #2: No

---

## [Editor Report · Acceptance letter]

12 Sep 2022

PONE-D-22-11055R1 

Understanding the long-term impact of flooding on the wellbeing of residents: a mixed methods study 

Dear Dr. Twiddy:

I'm pleased to inform you that your manuscript has been deemed suitable for publication in PLOS ONE. Congratulations! Your manuscript is now with our production department. 

Kind regards, 

on behalf of

Dr. Ali A. Weinstein 

Academic Editor

PLOS ONE